# A Critical Appraisal of the Physicochemical Properties and Biological Effects of Artificial Tear Ingredients and Formulations

**DOI:** 10.3390/ijms24032758

**Published:** 2023-02-01

**Authors:** Judy Weng, Michael K. Fink, Ajay Sharma

**Affiliations:** 1Chapman University School of Pharmacy, Chapman University, Irvine, CA 92618, USA; 2Department of Pathology, University of Colorado Anschutz Medical Campus, Denver, CO 80045, USA

**Keywords:** artificial tears, dry eye, CMC, hypromellose, polyols, osmoprotectants, hyaluronic acid, hydroxypropyl guar gum, preservatives

## Abstract

Dry eye disease is among the most prevalent diseases affecting the ocular surface. Artificial tears remain the cornerstone therapy for its management. There are currently a wide variety of marketed artificial tears available to choose from. These artificial tears differ significantly in their composition and formulation. This article reviews the physicochemical and biological properties of artificial tear components and how these characteristics determine their use and efficacy in the management of dry eye. Furthermore, this article also discusses the various formulations of artificial tears such as macro and nanoemulsion and the type of preservatives present in them.

## 1. Introduction

Tear Film & Ocular Surface Society Dry Eye Workshop II defines dry eye disease as a “multifactorial disease of the ocular surface characterized by a loss of tear film homeostasis accompanied by ocular symptoms, in which tear film instability and hyperosmolarity, ocular surface inflammation and damage, and neurosensory abnormalities play etiological roles” [1]. Due to its chronic and progressive nature, dry eye disease is emerging as among the most common reasons for ophthalmic outpatient visits. The prevalence of symptomatic dry eye with or without signs is estimated to range from 5% to 50% and the incidence reaching up to as high as 75% when classified primarily based on signs [2]. An estimated 16 million people in the United States suffer from dry eye disease [1]. Aging and female sex are significant risk factors since the prevalence of dry eye disease increases every five years in both men and women after the age of 50, and with a greater occurrence in women compared to men [2,3]. With the aging population and improved life expectancies, the prevalence of dry eye disease is expected to concomitantly increase, resulting in significant socio-economic consequences including loss of work productivity, decreased quality of life, and increased medical care spending in the management of dry eye [2]. Collectively, these factors account for an estimated USD 3.84 billion yearly in the United States, and this number is expected to rise to USD 6.2 billion in 2023, highlighting the increasing financial burden of dry eye disease on the health care system [2].

Characteristic symptoms of dry eye disease often present as ocular discomfort, which may include ocular dryness, itching, redness, pain, light sensitivity, foreign body sensation, and eye fatigue [2]. Commonly reported by patients of dry eye disease, ocular discomfort acts as a fundamental component in classifying the severity (mild, moderate, or severe) of dry eye and its monitoring, progression, and response to treatment. Based on etiology, dry eye disease has traditionally been classified into two major subtypes: aqueous tear deficiency due to inadequate tear production secondary to dysfunction of the lacrimal gland and evaporative dry eye due to increased tear film evaporation often resulting from meibomian gland dysfunction [1,4]. However, advances in current knowledge of dry eye disease pathophysiology have led to the reclassification of dry eye disease as a mixed continuum of both subtypes [1]. Furthermore, the use of some topical or systemic drugs, contact lens wear, and ophthalmic surgical procedures such as LASIK can cause iatrogenic dry eye. Current therapies for dry eye disease aim to restore tear film homeostasis and consist of a variety of pharmacological and non-pharmacological interventions ranging from over-the-counter artificial tears and warm compresses for the symptomatic management of mild dry eye, to prescription topical ocular anti-inflammatory medications for inflammation suppression associated with moderate to severe dry eye and in-clinic procedures targeting inspissated meibomian glands and tear conservation [5,6,7,8,9].

## 2. Artificial Tears in the Management of Dry Eye

The tear film is a thin fluid lining the ocular surface. With a thickness ranging from 2 to 5.5 μm and an osmolarity of approximately 300 mOsm/L, the tear film is a bilayer composed of an outer lipid and an aqueous and mucin layer, referred together as the mucoaqueous layer [10]. The lipid layer of the tear film is secreted by meibomian glands and plays a crucial role in lowering the surface tension of the tear film and resists tear evaporation by spreading across the ocular surface following a blink [10]. The aqueous component of the mucoaqueous layer is secreted by lacrimal glands and maintains ocular hydration. Ocular surface epithelial cells express membrane-tethered mucins 1, 4 and 16 that enhance lubrication and constitute a protective barrier function for the ocular surface [10,11]. Additionally, goblet cells and the lacrimal gland also release secreted mucins into tear film. Together, the components of the tear film bilayer work together to provide lubricity, hydration and protect the ocular surface against external pathogens and microbes. At rest, the high viscosity of the mucoaqueous layer of the tear film enhances tear film stability and increases resistance of tear film thinning [10,12]. However, high shear application by eyelids, due to blinking, results in a rapid drop of tear film viscosity as a protective mechanism to prevent damage to the underlying corneal epithelium [10,12]. Even though artificial tears are not meant to recapitulate natural tear functions, they remain the first line and mainstay treatment of dry eye disease. Artificial tears provide remarkable palliative and symptomatic relief in all stages of dry eye disease. Historically, artificial tears are topical eye formulations used to provide lubrication and maintain moisture on the ocular surface. In addition, artificial tears also play a crucial role in reducing ocular surface osmolarity and lowering of ocular surface inflammation through the dilution of inflammatory cytokines [6].

With the increasing prevalence of dry eye disease and the multitude of artificial tears currently available on the market, choosing the most appropriate artificial tear product for dry eye disease is crucial. Currently marketed artificial tear formulations contain a wide variety of biologically active ingredients, biologically relevant excipients, and preservatives that collectively contribute to their therapeutic benefit and in some cases, may cause undesirable effects. The goal of this article is to review the ingredients and formulations of artificial tears and their impact on the physicochemical properties, biological activity, and therapeutic use of artificial tears.

### 2.1. Physiochemical and Biological Properties of Active Ingredients in Artificial Tears

All artificial tears contain an active ingredient called demulcent (Table 1). The primary function of demulcents is to enhance viscosity and provide lubrication. Currently marketed artificial tears in the USA contain one of the six FDA-approved classes of demulcents: cellulose derivatives, gelatin, dextran 70, polyols, and polymers [13].

Carboxymethylcellulose (CMC) is a high-molecular-weight cellulose derivative with glucopyranose monomers backbone and carboxymethyl groups (-CH2-COOH) attached to the hydroxyl groups of glucopyranose. CMC has a high anionic charge with muco-adhesive and viscous properties, which allow it to have prolonged retention on the ocular surface to provide lubrication and wetting [14]. Additionally, CMC binds to the GLUT-1 glucose receptors on corneal epithelial cells, which further contributes to its prolonged ocular surface residence time [13,15]. CMC has good shear thinning properties, as demonstrated by neutron scattering and rheological studies [16]. It improves tear film integrity and provides corneal surface lubrication, making it an attractive demulcent that benefits patients suffering from mild to moderate dry eye disease [14]. CMC has been shown to enhance epithelial cell migration, improve wound healing, and has cytoprotective effects [15,17]. Therefore, CMC-based artificial tears have been used after laser in situ keratomileusis (LASIK) and contact lens insertion to accelerate postoperative ocular surface recovery and minimize lens-related epithelial cell injury. CMC is the primary active ingredient in Thera Tears^®^ and Refresh^®^ brand tears. The CMC concentration in these artificial tears ranges from 0.25% to 1%, with Thera Tears^®^ containing the lowest concentration at 0.25% [13]. At 1% concentration, CMC has a gel consistency with increased precorneal residence time, and notably enhances tear viscosity [13,18]. Although CMC leads to effective treatment of aqueous deficient dry eye in a dose-dependent manner, with the most improvement observed at 1% concentration, increased viscosity due to higher concentrations of CMC is often linked to the presence of eyelid debris and transient blur [15,19,20].

Hydroxypropyl methylcellulose (HPMC), also known as hypromellose, is a semisynthetic cellulose-based polysaccharide and due to its molecular size, it is less viscous than CMC [21,22]. Therefore, it is typically used in combination with dextran. Hypromellose restores the protective nature of the mucous layer of the tear film by crosslinking upon contact with the ocular surface due to a difference in pH, resulting in decreased tear clearance [23]. Thus, it enhances the moisturization and lubrication of the ocular surface [21]. Dextran 70 is a high-molecular-weight polysaccharide demulcent that has minimal viscosity enhancing effect and is used only in combination with other demulcents, like hypromellose [24]. Hypromellose, at 0.3% concentration, along with glycerin and dextran, is the main ingredient in the GenTeal^®^ brand of artificial tears. Hypromellose is a safe and effective ocular surface lubricant in managing dry eye disease, especially for patients with low-to-moderate dry eye symptoms [25]. Despite the differing properties of CMC and hypromellose, a recent clinical study demonstrates that both cellulose derivatives have a similar efficacy and safety profile in reducing symptoms of dry eye [22].

Polyols, also called polyhydric alcohols, are sugar-like hydrogenated carbohydrates. The three main polyol demulcents in currently marketed artificial tears are propylene glycol, polyethylene glycol, and glycerin with propylene glycol at 0.3% concentration and polyethylene glycol 400 at 0.4% concentration. Both polyethylene glycol and propylene glycol relieve ocular surface inflammation and irritation by forming a protective gel matrix over mucous membranes present on the ocular surface [26]. The Systane^®^ brand of artificial tears contains propylene glycol along with glycerin. Polyol demulcents are used at a concentration ranging from 0.2% to 1% in currently marketed artificial tear formulations, with glycerin often used at a concentration of 0.2% or 0.3% when combined with other lubricants.

### 2.2. Physiochemical and Biological Properties of other Biologically Active Ingredients in Artificial Tears

In addition to active ingredients, currently marketed artificial tears also contain a variety of biologically active components that act as osmoprotectants, humectants, mimic mucous or lipid layers (Table 2). These components especially in the artificial tears marketed in the USA are listed under the “inactive ingredients” but they have been reported to significantly contribute to the therapeutic benefits of artificial tears in dry eye disease.

Osmoprotectants are small, hydrophilic, osmotically active compounds known for their ability to modify cellular water uptake and protect the ocular surface against hyperosmolarity-induced injury [6,27]. Dry eye disease-induced hyperosmolarity of the tear film leads to apoptosis of corneal and conjunctival epithelial cells [14]. Osmoprotectants potentially safeguard the ocular surface epithelial cells from hyperosmotic stress-induced protein denaturation, cellular damage, and apoptosis. Additional studies have demonstrated that osmoprotectants regulate the autophagic process by reducing oxidative stress and matrix metalloproteinase synthase, further highlighting the multipronged mechanism of osmoprotectants in preventing dry eye disease-induced ocular surface epithelial cell damage [28,29,30]. Since osmoprotectants are internalized by the ocular surface epithelial cells, they have an increased duration of biological activity compared to demulcents that provide relief as long as they stay on the ocular surface before being washed off with tear clearance [31]. Based on their chemical structure, osmoprotectants include polyols (eg, erythritol, glycerin, sorbitol), methylamines (eg, betaine, glycine), and certain amino acids (eg, levocarnitine, taurine). Among these, the most common osmoprotectants in currently marked artificial tears include levocarnitine (L-carnitine), erythritol, and trehalose. L-carnitine is a small molecule present in prokaryotic and eukaryotic cells, including human corneal and conjunctival epithelia [13]. Erythritol is a four-carbon polyol used as a biological low-calorie sweetener [13]. In vitro studies have demonstrated that both L-carnitine and erythritol, when used alone or in combination, protect corneal epithelial cells from hyperosmolar stress by suppressing the inflammatory response and decreasing the concentration of intracellular inorganic salts [14,28]. In addition, clinical and in vivo studies have demonstrated that both L-carnitine and erythritol play critical roles in prevention and treatment of dry eye by significantly decreasing corneal staining, decreasing expression of inflammatory markers, reducing tear osmolarity, and providing symptomatic relief of dry eye [32,33,34,35]. Trehalose is a disaccharide that carries out its water regulation properties by forming a protective gel layer around organelles during cellular dehydration [14,28]. As an effective agent in treating moderate to severe dry eye disease, trehalose has been extensively studied. Clinical studies and in vivo animal studies highlight the cytoprotective role of trehalose for ocular surface epithelial cells. Specifically, studies have shown trehalose treatment in dry eye patients resulted in improvement of corneal fluorescein staining scores and prevented of corneal epithelial cells apoptosis [14].

Sodium hyaluronate and hyaluronic acid are the most common humectants that act as hygroscopic agents to enhance the retention of water on the ocular surface [6]. Hyaluronic acid, a high-molecular-weight naturally occurring glycosaminoglycan polysaccharide, is present in the aqueous humor, synovial fluid, and connective tissue [13]. In addition to its humectant properties, hyaluronic acid is also a viscosity-enhancing agent that increases tear film thickness and density [27]. As a humectant, hyaluronic acid retains water to the ocular surface by binding water multiple times the amount of its weight [13]. In vivo studies have demonstrated that hyaluronic acid, especially high-molecular-weight hyaluronic acid, accelerates epithelium wound healing after injury to the cornea, including alkali burn and corneal debridement abrasion likely by binding to CD44 receptor and shows significant improvement in dry eye mouse, rabbit and porcine models [14]. High-molecular-weight hyaluronic acid also reduces the incidence of ocular surface epithelial cell apoptosis and increase the tear break-up time [36,37,38]. Hyaluronic acid undergoes changes in viscosity due to its non-Newtonian nature upon the application of shear forces [39]. Specifically, rheological properties of hyaluronic acid containing artificial tears allow these formulations to undergo viscosity reduction during a blink, resulting in more even distribution of hyaluronic acid across the tear film [40]. Subsequently, at rest, higher viscosity is restored to allow prolonged residence time of hyaluronic acid on the ocular surface [40]. Studies have demonstrated that compared to carboxymethylcellulose and hydroxypropyl methylcellulose, hyaluronic acid had significantly better water retention properties and exhibited greater protection of corneal epithelial cells from desiccating stress [41]. Thus, artificial tears containing high-molecular-weight hyaluronic acid are suggested to be clinically superior in dry eye disease treatment compared to its low-molecular-weight counterpart [42].

Sodium hyaluronate, a semi-synthetic sodium salt derivative of hyaluronic acid with a smaller molecular weight, has mucoadhesive properties responsible for increasing corneal residence time in addition to its humectant features [13]. Structurally composed of carboxyl and hydroxyl groups, sodium hyaluronate is a more stable version of hyaluronic acid that absorbs large amounts of water to allow stabilization of the muco-aqueous layer of the tear film [43]. Artificial tears containing sodium hyaluronate maintain lubrication of the ocular surface by formation of a staggered, reticular structure on the eyeball to prolong water evaporation time [43]. Sodium hyaluronate also plays a major role in maintaining a protective barrier function on corneal epithelial cells due to its mucin-like molecular structure [43]. In addition, sodium hyaluronate has been shown to accelerate the repair of corneal and conjunctival epithelial cells, as it has similar extensibility characteristics as physiological tears [43]. Studies have also demonstrated that sodium hyaluronate triggers the migration, proliferation, and adhesion of corneal epithelial cells in dry eye disease patients, thus promoting corneal epithelial wound healing [44]. However, literature data suggest that sodium hyaluronate may show over-all a more restricted beneficial effects compared to high-molecular-weight hyaluronic acid. Only a handful of artificial tears in Europe contain high-molecular-weight hyaluronic acid and none of the currently marketed tears in the USA contain high-molecular-weight hyaluronic acid. There is currently very limited information on relative comparison of the efficacy of artificial tears containing hyaluronic acid compared to sodium hyaluronate in the treatment of dry eye. However, based on the preclinical data, it seems likely that hyaluronic acid containing tears will likely show a superior effective compared to the ones having sodium hyaluronate.

Hydroxypropyl guar gum, with its viscous protein-polymer characteristic, plays a key role in increasing the retention time of the aqueous layer of the tear film to mucins by increasing tear viscosity [6]. Derived from beans, hydroxypropyl guar gum is a semisynthetic gum containing numerous hydroxyl groups. It strengthens the attachment of the aqueous layer of tear film to mucins and promotes retention of other demulcents by forming a soft gel with increased viscosity at tear pH of 7.5, resulting in stabilization of the tear film [13]. In addition to its viscosity-enhancing properties, hydoxypropyl guar gum undergoes a solution-gelation transition that allows it to turn from a liquid to gel when mixed with tears [14]. This phenomenon happens due to the contact between hyroxypropyl guar gum with divalent ions and borates in tears and the formulation [14].

To mimic and maintain the integrity the tear film lipid layer, some artificial tears contain oils including castor oil, mineral oil, and flaxseed oil [6]. These lipid-containing artificial tears are formulated as emulsions and are classified based on the size of oil droplets as macroemulsions, microemulsions or nanoemulsions. Produced from castor beans, castor oil is primarily composed of ricinolein, the triglyceride of ricinoleic acid [45]. Castor oil has been shown to alleviate dry eye by increasing thickness of the lipid layer of the tear film [45,46]. Flaxseed oil is made up of a short chain of omega-3 fatty acids [47]. Mineral oil has been demonstrated to increase tear break up time and tear stability by thickening the lipid layer of the tear film. Overall, these oils in artificial tear formulations are proposed to be effective at reducing tear evaporation and are effective in managing evaporative dry eye including meibomian gland dysfunction. Multiple studies have shown that lipid-based drops are more effective than artificial tears containing only CMC and ypromellose [5,39]. The oil-containing artificial tears are likely to have a synergistic effect with humectants since a clinical study demonstrated that artificial tears containing both flaxseed oil and hyaluronic acid led to a better decrease in ocular surface proinflammatory markers and reduction in compromise of corneal epithelial barrier function compared to hyaluronic acid therapy alone [48].

Other inactive ingredients that are seen in artificial tears include carbomers and poloxamers. Carbomer, a polyacrylic acid, is a viscous synthetic polymer that is hydrophilic in nature and is used as an eye lubricant [49]. It forms a viscous gel and produces a transparent film on the ocular surface to maintain ocular surface lubrication [49]. Long precorneal resistance time is a unique feature of carbomer, allowing it to form a protective shield over damaged ocular surface epithelial cells for extended periods of time [49]. This enhanced protection from carbomer is thought to allow ocular surface cells to perform self-repair [49]. Poloxamer, a thermosensitive polymer, undergoes reversal thermal gelatin under specified temperatures and concentrations [50]. Traditionally, ophthalmic thermosetting gels are known to switch from a low-viscosity solution to a semi-solid gel when in contact with the temperature of the corneal surface [50]. Like carbomers, poloxamers also display an increase in pre-corneal residence time [50].

### 2.3. Effect of Formulations on the Biological Effects of Artificial Tears

In addition to the active and inactive ingredients, the type of formulation of artificial tears can also significantly impact their biological activity in the management of dry eye disease. Lipid-based eyedrops, usually available as oil-in-water emulsions, contain amphipathic lipids that improve tear film stability by reducing the interfacial tension between the lipid and aqueous layers of the ocular surface tear film [51]. While present in relatively minute concentrations in the tear film, phospholipids are important in maintaining stability of the tear film [51]. In addition, research has shown that phospholipid abnormalities may have potential etiological roles in the development of dry eye disease [51]. Thus, phospholipids are often included in the formulation of lipid-based eyedrops. Systane Balance^®^ and Systane Complete^®^ are examples of formulations that are emulsions with lipids and phospholipids. Systane Balance^®^ is formulated as a microemulsion with mineral oils and polar phospholipid surfactants, which are responsible for its milky white appearance [51]. This formulation allows Systane Balance^®^ to reduce tear evaporation from the ocular surface [26]. Systane Complete^®^ is a nanodrop formulation with a patented LipiTech™ system that has otherwise identical composition to that Systane Balance^®^ [13] but contains a 3-fold higher concentration of hydroxypropyl guar gum. Due to nanodrop size, both preclinical and clinical studies show that Systane Balance^®^ artificial tears are more efficacious in the management of dry eye as compared to Systane Complete^®^ allowing tiny, nano-sized droplets to form a protective matrix across the entire ocular surface [52]. These lipid-containing emulsions artificial tear drops are gaining popularity and becoming more widespread due to their role in preventing tear evaporation and restoring the lipid layer of the tear film [5]. This is crucial, as many dry eye patients suffer from meibomian gland dysfunction, leading to depletion of lipid layer of the tear film. Due to their droplet size, these lipid formulations are visually cloudy and can induce visual blur when applied topically [5]. Emulsion artificial tear formulations should be shaken or inverted to enhance uniformity prior to application [5]. This limitation has been addressed by meta-stable emulsions have been employed by many commercial products, like Systane^®^ brand artificial tears [5].

### 2.4. Impact of Preservatives on the Use of Artificial Tears

The use of chemical preservatives to inhibit or minimize the bacterial load in artificial tears is commonplace to lengthen the shelf life of multidose vials. Alternatively, single-dose units meant for one time application do not require the addition of preservatives. Benzalkonium chloride (BAK), polyquaternium-1 (Polyquad), Purite and sodium perborate are the most common preservatives in currently marketed artificial tears. Among current brand name artificial tears, Soothe^®^, Akron^®^, and Murine^®^ brand tears use benzalkonium chloride as a preservative. Refresh^®^ brand artificial tears utilize purite, whereas sodium perorate is used in TheraTears^®^ and polyquaternium-1 is generally found in GenTeal^®^ and Systane^®^ branded tears. Although benzalkonium chloride (BAK) is among the least expensive preservatives, its use is associated with significant ocular surface toxicity. BAK is an amphiphilic quaternary ammonium compound that kills bacteria by surfactant action on their plasma membrane [53]. However, mammalian cells, including the corneal and conjunctival epithelial cells, can also absorb and be impacted by the surfactant action of BAK [53]. In vitro studies using corneal and conjunctival epithelial cells demonstrate that BAK decreases ocular surface epithelial cell viability and increases the release of pro-inflammatory mediators, illustrating the direct toxicity BAK has on the ocular surface epithelial cells [54,55,56,57,58]. In addition, in vivo studies demonstrate that topical application of BAK in rabbits and rodents leads to tear film abnormalities, reduction in mucins and goblet cell density, corneal epithelial desquamation, changes in corneal innervation and an influx of inflammatory cells [59,60,61,62,63,64,65]. In line with these preclinical studies, the prolonged use of BAK-containing eye drops in glaucoma patients has also been shown to cause severe damage to the ocular surface epithelial cells and corneal nerves [53]. Furthermore, glaucoma patients using BAK-containing eye drops have been reported to suffer from reduced tear film and tear breakup time, as well as increased tear osmolarity [53,66]. The detrimental effects associated with BAK are cumulative and increase in severity due to repeated exposure and at higher concentrations [53]. Due to the observed toxic effects of benzalkonium chloride on exacerbation of ocular surface damage and further worsening of the inflammatory response highlighted in in vitro, in vivo, and clinical studies, the use of BAK containing artificial tears should be avoided in dry eye patients, especially those with moderate to severe dry eye.

Polyquaternium (Polyquad), like BAK, is a quaternary ammonium compound with bactericidal activity due to its surfactant action. With a molecular size approximately 27-fold larger than BAK, Polyquad is a comparatively less toxic alternative, as its large molecular size reduces its ability to penetrate and disrupt mammalian cells including ocular surface epithelial cells [53,67]. Thus, while maintaining its antibacterial activity, Polyquad has not been shown to cause damage to mammalian cells [53]. Several in vitro and in vivo studies comparing the ocular surface toxicity of polyquaternium and benzalkonium chloride show that Polyquad has lower cytotoxicity, is better tolerated upon topical instillation and it is remarkably safer than benzalkonium chloride [68,69,70,71].

Sodium perborate, also known as Dequest^®^, is an oxidative, vanishing type of preservative that is converted to hydrogen peroxide in solution and causes the bactericidal effect by oxidative stress-mediated denaturation of bacterial proteins [53]. Sodium perborate is decomposed to oxygen and water by the enzyme catalase upon instillation into the eye [53]. While its use is currently limited in marked artificial tears, sodium perborate is still an effective preservative due to its antimicrobial activity and minimal impact on the ocular surface [53]. Multiple in vitro studies have demonstrated that while sodium perborate is less toxic than benzalkonium chloride, it is not completely innocuous [72,73]. Stabilized oxychloro complex (SOC), also known as Purite^®^, is a vanishing type of preservative that generates chlorine dioxide free radicals responsible for its oxidizing antimicrobial activity in solution and is converted to oxygen, water, sodium and chloride ions upon administration to the eye [74]. Multiple in vivo rabbit studies comparing the use of SOC and BAK-preserved glaucoma medications and artificial tears demonstrate that SOC is a superior alternative as it results in significantly less corneal damage, decreased corneal epithelium inflammatory cells, and better tolerability compared to BAK-preserved formulations [75,76].

As an alternative to preservative containing artificial tears is to use a preservative-free eye drop [53]. Single-dose units with a twist off cap containing 0.1 and 1.0 mL of fluid are the most common alternatives. They are intended to be discarded after a single use and thus are significantly more expensive treatment option, costing up to 5–10-fold higher compared to multidose vial. The use of membranes with antimicrobial properties or bottle with microbial filters are emerging as alternatives to the use preservatives yet maintaining sterility in multidose vials. Many such systems like ABAK^®^ bottle, Clear Eyes and the hydraSENSE^®^ delivery system and COMOD^®^ dosage system are currently being used especially in European market [53].

## 3. Conclusions

In summary, currently marketed artificial tears contain a wide variety of ingredients that can affect their biological properties and therapeutic use, thus playing an important role in selecting an optimal artificial tear in the management of dry eye disease. Multiple recent articles have reviewed the literature that have compared the clinical efficacy of various artificial tears with these different ingredients [13,14,25,77,78,79]. These articles have come up with some general guidelines for selecting optimal artificial tears depending on the severity and type of dry eye disease. Further, it is very challenging to make conclusive recommendation for the use of artificial tears based on these clinical trials since there is variability in the study design, tested clinical end points, patients’ heterogeneity, and sponsor bias. The aim of the current article was to review the composition and formulation of artificial tears and their impact on the physicochemical and biological effects of tears on the ocular surface. Reviewing these artificial tears for recommendation of their clinical use is beyond the scope of this article. Thus, readers seeking such information should refer to other excellent reviews that focus on the clinical use of artificial tears [13,14,25,77,78,79]. However, some general guidelines are that lipid-based artificial tears and tears containing sodium hyaluronate or hyaluronic acid seem to show better efficacy especially in evaporative dry eye. Furthermore, having other biological active ingredients, such as hydroxypropyl guar gum or osmoprotectants, further enhances the efficacy of artificial tears. Tears containing demulcents only, such as CMC or hypromellose alone, may be more suitable for use in mild cases of dry eye. Finally, artificial tears are a dynamic and ever evolving field, as is highlighted by the recent European approval of a lipid-only formulation containing perfluorohexyloctane. Perfluorohexyloctane (F6H8) is a non-aqueous alkane liquid that is applied topically and mimics the lipid layer of the tear film, leading to reduction of tear evaporation and increase in tear film stability [80]. Thus, it is an effective form of treatment for patients suffering from evaporative forms of dry eye disease secondary to meibomian gland dysfunction [80]. Due to its physiochemical properties, it does not require the use of a preservative, making it an attractive alternative to many currently marked artificial tears [80].

## Figures and Tables

**Table 1 ijms-24-02758-t001:** List of active ingredients in artificial tears.

Active Drug	Physiochemical Property	Market Brands
Carboxymethylcellulose (CMC)	Viscosity enhancing, mucomimetic	Thera Tears^®^, Thera Tears Extra^®^, Refresh Tears^®^, Refresh Plus^®^, Refresh Optive^®^, Refresh Repair^®^, Refresh Liquigel^®^, Refresh Celluvisc^®^
Hydroxypropylmethylcellulose	Mucoadhesive, mucomimetic and water retention when used with glycerin and dextran	GenTeal Mild^®^, GenTeal Moderate^®^, GenTeal Severe^®^
Propylene glycol, polyethylene glycol, glycerin	Water retention, restoration of lipid layer of tear film	Systane Balance^®^, Systane Complete^®^, Systane Original^®^, Systane Ultra^®^, Systane Gel Drops^®^, Systane Hydration^®^, Oasis Tears^®^, Blink Tears^®^, Soothe^®^

**Table 2 ijms-24-02758-t002:** List of other biologically active ingredients in artificial tears.

Active Drug	Biological Property	Market Brands
Levocarnitine (L-carnitine)	Osmoprotectant	Refresh Optive^®^, Refresh Optive Advanced^®^, Refresh Optive Mega-3^®^, Refresh Optive Gel^®^
Erythritol	Osmoprotectant	Refresh Optive^®^, Refresh Repair^®^, Refresh Optive Advanced^®^, Refresh Optive Mega-3^®^, Refresh Optive Gel^®^
Trehalose	Osmoprotectant	Thera Tears Extra^®^, Refresh Optive Mega-3^®^
Hyaluronic acid and sodium hyaluronate acid	Water retention, viscosity enhancing, mucomimetic	Refresh Repair^®^, Systane Hydration^®^, Oasis Tears^®^, Oasis Tears Plus^®^, Blink Tears^®^
Hydroxypropyl guar gum	Water retention, increases tear viscosity, mucomimetic	Systane Balance^®^, Systane Complete^®^, Systane Original^®^, Systane Ultra^®^, Systane Gel Drops^®^, Systane Hydration^®^
Castor oil	Mimics lipid layer of tear film, reduces evaporation	Restasis^®^, Refresh Optive Mega-3^®^, Refresh Optive Advanced^®^
Mineral oil	Mimics lipid layer of tear film, reduces evaporation	Systane Balance^®^, Systane Complete^®^
Flaxseed oil	Mimics lipid layer of tear film, reduces evaporation	Refresh Optive Mega-3^®^

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
