# Peer review of "A Critical Appraisal of the Physicochemical Properties and Biological Effects of Artificial Tear Ingredients and Formulations"

_ijms, 2023, doi:10.3390/ijms24032758_

Round 1

Reviewer 1 Report

Authors have done a a very thorough job at reviewing the physicochemical properties of artificial tears which will be a great aid to any clinician managing dry eye disease. 

The introduction is sound and highlights the burden of dry eye disease. However, with the theme of the article, it would be appropriate to add the physicochemical properties of the human tear film so as to help readers understand and draw comparisons mentioned in the rest of the text.

A few minor edits:

Line 100 and 101 needs references.

Author Response

  1. Authors have done a very thorough job at reviewing the physicochemical properties of artificial tears which will be a great aid to any clinician managing dry eye disease. 

Thank you so much for the encouraging feedback.

  1. The introduction is sound and highlights the burden of dry eye disease. However, with the theme of the article, it would be appropriate to add the physicochemical properties of the human tear film so as to help readers understand and draw comparisons mentioned in the rest of the text.

Thank you so much for this great suggestion. As per your suggestion, we have added a new paragraph that describes the composition and the physicochemical properties of the tear film (page 2, line 57-72)

  1. Line 100 and 101 needs references.

As suggested, we have added the references to these two lines (Ref # 21,22, line122)

Reviewer 2 Report

Congratulations! Your manuscript is so interesting. 

I do not have many recommendations. I have seen that you repeat many cites (for instant 1, 5 or 11)  time after time . Could it be possible to replace them?

Author Response

  1. I do not have many recommendations. I have seen that you repeat many cites (for instant 1, 5 or 11) time after time. Could it be possible to replace them?

Thank you so much for this suggestion. As suggested, we have replaced some these references, however these are some of the comprehensive review articles in the field including a 10 year updated reports by published by TFOS (ref #1 and #5), thus it necessitated their citation at multiple places.